# Colour Stability of Two Commercially Available Maxillofacial Prosthetic Elastomers after Outdoor Weathering in Al Jouf Province

**DOI:** 10.3390/ma16124331

**Published:** 2023-06-12

**Authors:** Mahmoud Gamal Salloum, Kiran Kumar Ganji, Ali Mohammed Aldajani, Shital Sonune

**Affiliations:** 1Department of Substitutive Dental Sciences, College of Dentistry & Pharmacy, Buraydah Private College, Buraydah 51418, Saudi Arabia; mahmoud.salloum@bpc.edu.sa; 2Department of Preventive Dentistry, College of Dentistry, Jouf University, Sakaka 72345, Saudi Arabia; 3Department of Periodontics & Implantology, Sharad Pawar Dental College & Hospital, Datta Meghe Institute of Higher Education & Research, Sawangi (Meghe), Wardha 442107, India; 4Department of Prosthetic Dental Sciences, College of Dentistry, Jouf University, Sakaka 72345, Saudi Arabia; dr.alaldajani@jodent.org (A.M.A.); dr.shital.sonune@jodent.org (S.S.)

**Keywords:** colour stability, elastomers, weathering, maxillofacial, dentistry, maxillofacial surgery

## Abstract

Facial prostheses are created from special elastomers modified for their specific physical and mechanical properties; however, they also show two common major clinical problems: gradual discolouration of the prosthesis over time in service environment and deterioration of static, dynamic, and physical properties. As a result of external environmental factors, facial prostheses may become discoloured and discolour by changing colour from intrinsic and extrinsic colouring, and this is associated with the intrinsic colour stability of elastomers and colourants. Thus, in this in vitro study, a comparative evaluation of the effect of outdoor weathering on the colour stability of A-103 and A-2000 room-temperature vulcanised silicones used for maxillofacial prosthesis was conducted. To accomplish this study, a total of 80 samples were fabricated, 40 samples of each material were grouped as clear (20) and pigmented (20). These samples were mounted on wooden board and the assembly was placed on the roof of the dental school from October 2021 to March 2022. To maximise the amount of sunlight on the specimens, the exposure rack was set on five 68° angles from horizontal and also to prevent standing water. The specimens were left uncovered during exposure. The testing of samples was conducted with the help of a spectrophotometer. The colour values were recorded in the CIELAB colour system. It describes the three colour coordinates (colour values) x, y, and z in three new reference values of L, a, and b, aiding in numerically classifying colour differences. After 2, 4, and 6 months of weathering, testing was conducted using a spectrophotometer and the colour change (ΔE) was calculated. The A-103 RTV silicone group with pigmentation showed the maximum change in colour after six months of environmental conditioning. The data for colour difference within groups were analysed using a one-way ANOVA test. Tukey’s post hoc test assessed the pairwise mean comparison’s contribution to the overall significant difference. The nonpigmented A-2000 RTV silicone group showed the maximum change in colour after six months of environmental conditioning. After 2, 4, and 6 months of environmental conditioning, pigmented A-2000 RTV silicone showed better colour stability than A-103 RTV silicone. The patients requiring facial prosthesis do need to work on outdoor fields, and thus weathering will have deleterious effects on such prosthesis. Hence, the selection of appropriate silicone material with respect to the Al Jouf province region is crucial, which includes economic, durable, and colour stability.

## 1. Introduction

The demand for maxillofacial prostheses for rehabilitation of patients with congenital and acquired defects increased in recent years [1]. Maxillofacial prosthetics is a branch of prosthodontics concerned with the restoration and/or replacement of stomatognathic and craniofacial structures with prostheses that do not require removal on a daily or regular basis [2,3]. Restoration of aesthetics in patients with gross defects of the face and head is often a valuable and sometimes dramatic service provided by maxillofacial prosthodontists. The replacement of missing parts, such as the nose, eye, and ear, or the fabrication of a prosthesis to rebuild facial or cranial contours, require the utmost clinical skill and utilisation of the materials that are available. The five main objectives of maxillofacial prosthodontics are: 1. restoration of aesthetic or cosmetic appearance of the patient; 2. restoration of function; 3. protection of the tissues; 4. providing therapeutic or healing effects; and 5. psychological satisfaction. External maxillofacial prostheses were used for several decades for the anatomical, functional, or cosmetic restoration of the maxilla, mandible, or face that is missing or damaged by disease, accident, or congenital malformation [4]. Massive destruction of the facial structures often results in large and unsightly defects. Defects that cannot be successfully repaired by reconstructive surgical procedures may be remedied using facial prosthesis [5] to shield exposed tissues, cover defect cavities, and restore physical appearance [6].

Over the past few decades, significant developments in maxillofacial surgery dramatically increased the demand for more applicable materials [7]. The optimal maxillofacial materials should have optical and mechanical properties comparable to those of the replaced human tissue, and the materials should maintain their properties during service [8]. Many researchers previously reviewed the characteristics of an ideal material for facial prostheses; however, an ideal prosthetic material is yet to be produced. Among the various required mechanical and performance characteristics of prosthetic materials, the ones that attract most attentions are sufficient hardness, high tensile strength, colour stability, and resistance to ultraviolet light [9,10]. Commercial- and experimental-grade polymers, including poly(methyl methacrylate), poly(vinyl chloride), chlorinated polyethylene, polyurethanes, and silicones, were widely employed in facial prostheses in recent decades because of their improved physical properties [11]. Polydimethylsiloxanes are popular materials for fabricating maxillofacial prostheses, owing to the ease of fabrication. Chlorinated polyethylene is a class of thermoplastic elastomers with enhanced properties developed as maxillofacial materials for facial prostheses [12]. Although polymers are versatile materials, their performance is far from ideal, exhibiting disadvantages such as reduced thermal stability and diminutive resistance to solar radiation.

Weathering of polymers can induce changes in their physical and chemical characteristics, resulting in significant alterations to their mechanical and optical properties [13,14,15]. Common environmental factors that cause polymer degradation include sunlight, temperature, moisture, wind, dust, and pollutants. More precisely, the deterioration of the material could be a result of a photo-oxidative attack, which is the combined action of oxygen and sunlight on the chemical structure of the material [16]. The ability to resist colour alteration when exposed to sunlight over an extended period is one of the most desirable performance characteristics of ideal facial prosthetic elastomer [8]. The discolouration of a facial prosthesis can be caused by the degradation of the intrinsic colour stability of elastomers and colourants (pigments, flocking) as a result of the external environmental factors. Irrespective of the type of elastomer used in the fabrication of the facial prosthesis, the service life is generally between 6 months and 2 years, with an average of 10–12 months [17]. For this reason, studies on the discolouration of pigmented and nonpigmented facial elastomers after exposure to sunlight, air pollution, ambient temperature, and humidity for extended periods are essential for understanding the failure mechanism in a service environment. However, little information is available in the literature on the colour stability of facial materials during outdoor weathering. Therefore, the aim of this study was to evaluate and compare the effects of outdoor weathering conditions, especially in the Jouf Province (Kingdom of Saudi Arabia), on the colour stability of two commercially available room-temperature vulcanised silicones used in maxillofacial prostheses.

## 2. Materials and Methods

### 2.1. Study Setting and Materials

The study was conducted at the College of Dentistry, Jouf University, Sakaka, Al Jouf, Saudi Arabia. Two room-temperature vulcanising maxillofacial silicone materials were tested in this study: A-103 (MED-4210, Factor II Inc.; Lakeside, AZ, USA) and A-2000 (Factor II Inc.; Lakeside, AZ, USA), with an intrinsic pigment, FI-SK49 in Almond (Factor II Inc.; Lakeside, AZ, USA). The master mould (Jeltrate Plus, Dentsply Sirona, York, PA, USA) was fabricated using a 30 mm diameter and 3 mm thickness disc made of self-curing acrylic resin (resin repair material, BMS Dental, Capannoli, Italy). Five putty indices were prepared using this mould (Jeltrate Plus, Dentsply Sirona, York, PA, USA). Hard wax (GEO Natural, Renfert, Hilzingen, Germany) was used to prepare twenty-one discs using the putty index. High-strength dental stones (Hera, Kulzer, Leipziger Strabe 2, Germany) were mixed according to the manufacturer’s instructions and poured into square disposable plastic containers. Wax discs were then placed on the dental stone (Hera, Kulzer, Leipziger Strabe 2, Germany). Once the dental stone (Hera, Kulzer, Leipziger Strabe 2, Germany) mould was hardened and set completely, it was placed in a hot water bath (Unident, Västerhavsvägen 2, Sweden), and thereafter held under running hot water to ensure the complete removal of the wax. Prior to preparing the samples, it was ensured that the dental stone mould was completely dry. To perform power analysis and sample size computation, we used the G*Power software (version 3.1.9.7, University of Düsseldorf, North Rhine-Westphalia, Germany). With a power of 80% and significance level of 5%, we performed power analysis to estimate the minimum sample size required to detect significant differences among the four groups in our study, which was set to be 80, with a moderate effect size (Cohen’s d = 0.5) and a standard deviation of 10 units based on the literature and preliminary results. Within the resource and timeline constraints, the sample size was deemed feasible and practical.

### 2.2. Preparation of Clear and Pigmented Samples

For the A-103 samples, to prepare 20 clear samples, 60 g of Part A (silicone polymer base) and 6 g of Part B (crosslinking agent) were measured and vacuum mixed at 28 inch Hg for 30 min [18] to obtain a uniform mixture without air bubbles. Thereafter, the mixed material was poured into the stone mould coated with a thin layer of separating medium. Gentle tapping was performed to ensure uniform filling of the mould. According to the manufacturer’s instructions, the mould was maintained at room temperature for 24 h for complete curing. Finally, disc-shaped clear silicone samples of 30 mm in diameter and 3 mm in thickness were prepared. The same procedure was used for the preparation of the A-103 pigmented samples with addition of the pigment.

For the preparation of A-2000 samples, Part A and Part B were mixed at a 1:1 ratio at 23–250 °C with 50 ± 10% humidity. To prepare 20 clear samples, 40 g of Part A and 40 g of Part B were measured and vacuum mixed at 28 inch Hg for 10 min. As per the manufacturer’s instructions, the mould was initially kept in a dry heat oven at room temperature for 12 h, before elevating the temperature to 73 °C and keeping the samples in the oven overnight for complete curing. Finally, disc-shaped clear silicone samples with a diameter of 30 mm and thickness of 3 mm were prepared and stored in an airtight container in the dark. The same procedure was used for the preparation of the A-2000 pigmented samples with the addition of intrinsic pigment. All the prepared samples were categorised into Groups I and II (Figure 1).

### 2.3. Colour Change Evaluation

The prepared samples were mounted on four wooden boards that were 15 in long, 12 in wide, and 1.5 in thick. On each board, the samples were arranged in four equally spaced rows, secured using stainless-steel ligature wire, and numbered from 1 to 20 in each group. These numbers were written below each sample on the wooden board. The CIELAB colour scheme was used to record the values, which have uniform colour space with lightness coordinates; that is, white-black (L*), redness–greenness (a*), and yellowness–blueness (b*) [19]. The L*a*b* values of each specimen were measured at baseline (before being mounted to the boards) and after 2, 4, and 6 months of outdoor exposure using a spectrophotometer (SpectroShade Micro II, MHT Optic Research AG, Zurich, Switzerland) with a standard white background. Before measuring the shade, a white–green calibration was performed, as suggested by the manufacturer. Positioning was employed to ensure that the readings for each specimen were obtained at the same point. Values were recorded using the CIELAB colour system [20].

### 2.4. Environmental Exposure

The atmospheric environmental exposure of the specimens followed the American Society for Testing and Materials Designation G7-83. The wooden boards with mounted samples were placed on an exposure rack. To avoid standing water and maximise the amount of sunlight irradiation on the specimen, the exposure rack was angled at 5° from the horizontal level. The experiment was conducted on the roof of a dental school from October 2021 to March 2022. The specimens were uncovered and exposed to the environment, and the detailed weathering conditions are provided (Table 1).

### 2.5. Testing of Samples after Environmental Exposure

The specimens were placed for 10 min in distilled water in an ultrasonic cleaner and wiped dry before measurement. Samples were tested in the same manner using a spectrophotometer at intervals of 2, 4, and 6 months. The colour change (ΔE) was calculated using the following equation: $\Delta E={\left({\left[\Delta L^{*}\right]}^{2}+{\left[\Delta a^{*}\right]}^{2}+{\left[\Delta b^{*}\right]}^{2}\right)}^{1/2}$, where ΔE is the change in colour, ΔL* is the change in L* between the interval of interest and baseline, Δa* is the change in a* between the interval of interest and baseline, and Δb* is the change in b* between the interval of interest and baseline.

### 2.6. Statistical Analysis

Statistical analysis was performed using SPSS software (IBM corporation, Version 22, Chicago, Illinois, NY, USA). The mean and standard deviation values of ΔE were used as primary outcomes to assess the colour difference among the subgroups. The Shapiro–Wilk test for ΔE values revealed that the data were normally distributed (*p* > 0.05). Parametric tests were performed to assess the data. The data of ΔE related to colour differences were analysed using one-way analysis of a variance test (ANOVA) followed by Tukey’s post hoc test to assess pairwise mean comparisons contributing to the overall significant difference. For all evaluations, α < 0.05 was considered as statistically significant.

## 3. Results

In total, 80 samples were analysed for colour differences after weathering for periods of 2, 4, and 6 months, which were distributed to Group I Control, Group I Pigmented, Group II Control, and Group II Pigmented. A colour difference formula was used based on the CIE L*a*b* system. Values of ΔE* ≤ 3 were considered clinically acceptable and values greater than 3 were considered clinically indicative of colour differences, as suggested by Fontes et al. [21]. The ΔE values of Group I Control are presented using box and whisker plots, showing the lowest, highest, median, first, and third quartiles in Figure 2. In Group I Control, the ΔE values ranged from 1.8 to 5.8, 3.6 to 6.3, and 4 to 7, evaluated after 2, 4, and 6 months, respectively. The ΔE values of Group I Pigmented ranged from 4.5 to 7.4, 4.4 to 8.6, and 4.5 to 8.7, evaluated after 2, 4, and 6 months, respectively (Figure 3). The ΔE values of Group II Control ranged from 1.5 to 7.0, 2.9 to 7.0, and 3.4 to 7.0, evaluated after 2, 4, and 6 months, respectively (Figure 4). The ΔE values of Group II Pigmented ranged from 1.2 to 4.1, 1.4 to 4.1, and 1.7 to 4.7, evaluated after 2, 4, and 6 months, respectively (Figure 5).

The mean ΔE values of Group 1 Control, Group 1 Pigmented, Group II Control, and Group II Pigmented are presented in Table 2.

Evaluation of samples at 2 months: among all the groups the mean ΔE value of Group II Pigmented was the lowest (2.86) and clinically acceptable, whereas the mean ΔE values of other groups ranged from 4.19 to 6.23, demonstrating colour difference clinically.

Evaluation of samples at 4 months: Among all the groups the mean ΔE value of Group II Pigmented was the lowest (3.09) and clinically unacceptable. The mean ΔE values of other groups ranged from 4.92 to 6.54, demonstrating colour difference clinically.

Evaluation of samples at 6 months:all the groups demonstrated colour difference as the mean ΔE values were greater than 3 and ranged from 3.47 to 6.68.

Overall Group I Pigmented presented the highest mean ΔE values upon evaluating at 2, 4, and 6 months, suggesting that the elastomeric material in this group was clinically unacceptable for colour changes.

Figure 6 and Figure 7 represent the graphical comparison of mean ΔE values after environmental exposure for 2, 4, and 6 months. Marginal differences in the mean ΔE values were observed between Group I Control and Group II Control, and a major difference in the mean ΔE values was observed between Group I Pigmented and Group II Pigmented.

A one-way ANOVA was conducted with all groups as the independent variable and mean ΔE values measured at 2, 4, and 6 months as the dependent variable. Results show a significant difference (*p* < 0.05) between the groups (Group 1 Control, Group 1 Pigmented, Group II Control, and Group II Pigmented) on ΔE values (Table 2) at 2, 4, and 6 months. Pairwise comparison demonstrated a statistically significant difference (*p* < 0.05) for difference in ΔE values at 2, 4, and 6 months for Group 1 Control, Group 1 Pigmented, Group II Control, and Group II Pigmented (Table 3). Post hoc comparison using Bonferroni adjustments revealed that there was a statistically significant difference in mean ΔE values between Group I Control and Group I Pigmented (*p* < 0.05), Group I Control and Group II Pigmented (*p* < 0.05), and Group I Pigmented and Group II Pigmented (*p* < 0.05). There was a statistically insignificant difference in mean ΔE values between Group I Control and Group II Control (*p* = 0.978).

## 4. Discussion

Maxillofacial prostheses require frequent replacement because they become discoloured or rapidly degrade in service due to degradation of the base elastomer [22]. In 1980, Chalian, Drane, and Standish described the important criteria for maxillofacial materials as ease of application and retention, colour stability, durability, lack of toxicity, strong peripherals, translucency, ease of cleaning, lightweight, ease of fabrication, and physical and chemical inertness [23]. In maxillofacial silicone prostheses, colour deterioration cannot be attributed to a single factor or to aging alone. Various factors, such as exposure to the environment, humidity, UV radiation, air pollution, and facial secretions, as well as the disinfection methods, contribute to its occurrence. In addition to these external factors, internal factors also play a significant role. For example, the silicone prostheses can be affected by the silicone composition, degree of crosslinking, curing method, and the extrinsic and intrinsic stains. Al-Harbi et al. investigated the effects of outdoor weathering on the mechanical properties and colour stability of three silicones in a hot and humid climate, and concluded that outdoor weathering adversely affected both properties [24]. Several silicone elastomers, including M511, were evaluated using simulated skin secretions to determine their effects on the mechanical properties and colour stability [15]. Material colour stability refers to whether the materials maintain their colour over time in a certain environment. The Munsell and CIE L*a*b* colour systems are used to assess chromatic differences, as recommended by the American Dental Association, to determine colour changes.

Developing a silicone elastomer that meets the criteria for an ideal material cannot be overstated. Maxillofacial prostheses with highly aesthetic and structurally durable qualities can have a dramatic effect on a person’s quality of life. The psychological influence of head and neck defects, such as shame, anxiety, self-esteem reduction, and feelings of inferiority, may adversely affect the psychological well-being of patients if these conditions are left unrehabilitated [25]. Ideally, materials employed for the fabrication of maxillofacial prostheses should possess high colour stability upon outdoor weathering in order to be aesthetically acceptable and serviceable for a long time before they need to be remade. Therefore, the current study was designed to evaluate and compare the effects of outdoor weathering conditions on the colour stability of two commercially available room-temperature vulcanised silicones (A-103 and A-2000) used for maxillofacial prostheses.

The choice of the correct elastomer was crucial. Montgomery and Kiat-Amnuay recently selected two silicone elastomers from a survey to test room-temperature vulcanised silicone elastomers. To identify the most commonly used silicone elastomers in maxillofacial prosthetics, 260 members of the American Anaplastology Association and American Academy of Maxillofacial Prosthetics (AAMP) completed a survey. Based on the responses of 43 respondents, the authors found that silicones developed by Factor II (Lakeside, AZ, USA) were preferred and the most frequently used material was A-2186 (Factor II). A-2186, A-2186F, A-2000 (Factor II), and MDX4-4210 with catalyst A-103 (Dow Corning Corp., Midland, MI, USA) are among the most common silicone compounds [26]. Materials were selected based on this survey to provide clinically relevant information. Therefore, Factor II A-2000 was used in this study. The material A-2186F (Factor II) was excluded from this study because its formulation is similar to that of A-2186, as it was developed as a faster polymerising version of A-2186 with a higher platinum content. A-103 (Factor II) was selected as the final alternative to MDX4-4210 (Dow Corning Corp., Midland, MI, USA). Materials A-103 and MDX4-4210 exhibited the same formulation; a dimethylsiloxane polymer, reinforcing silica, and platinum catalysts were the main components of these materials. Additionally, both curing agents contain a dimethylsiloxane polymer, an inhibitor, and a siloxane cross-linker. Biocompatibility tests performed on medical-grade elastomers and silicone rubber yielded comparable results [27,28]. The elastomer production lots were tested for cytotoxic effects using a tissue culture test (the direct contact method) [27]. As both materials were indistinguishable in their formulation and MDX4-4210 was not available, A-103 was selected for testing as an alternative. Additionally, in the current study, the functional intrinsic skin colour (Factor II, Lakeside, AZ, USA) was carefully chosen to colourise the silicone materials. This concept was based on a survey conducted by Montgomery and Kiat-Amnuay, who evaluated the most commonly used pigments as intrinsic opacifiers in the fabrication of maxillofacial prostheses in comparison with oil and dry earth pigments [26,28].

The results of this study demonstrate that A-2000 combined with Factor II (Group II Pigmented) had a lesser amount of colour change than the A-103 MED-4120 platinum silicone specimens exposed to outdoor weathering. Because polymeric biomaterials cannot withstand high thermal changes and show little resistance to sunlight, they are susceptible to deterioration upon exposure to weather conditions [29]. In addition to solar radiation and temperature, the chemical nature of silicone elastomers can be altered by environmental factors such as sun exposure, temperature, and humidity. Simulating real outdoor conditions can determine the performance of silicone elastomers under extraoral conditions [30]. Because silicone prostheses react to solar radiation, moisture, and temperature, the colour of silicone prostheses can change over time, causing patient dissatisfaction. In agreement with other studies, [14,23] the present study revealed that all groups exposed to outdoor weathering exhibited significant colour changes as expressed by ΔE (*p* < 0.05) by time of evaluation (2, 4, and 6 months). Visually perceptible and clinically unacceptable colour changes occurred for the A-103, A-103 pigmented, and A-2000 elastomers (ΔE > 3), whereas only the A-2000 pigmented elastomers were clinically acceptable (ΔE < 3) in this study. During outdoor weathering, the four silicone elastomers exhibited visually detectable colour differences (ΔE*). Farah et al. [31] studied the effects of dark storage, accelerated aging in a weathering laboratory, and outdoor weathering on the colour stability of nonpigmented and pigmented maxillofacial silicone. According to the authors, the colour difference (ΔE*) ranges from 2 to 4.86. We observed colour differences ranging from 2.8 to 6.6. This six-month exposure period should be interpreted carefully to correlate our results with the actual clinical use of facial prostheses. Interestingly, there were significant differences in ∆E values measured at 2, 4, and 6 months, indicating a positive influence of weathering on elastomers. It is unlikely that a patient wearing a facial prosthesis will spend an entire year outdoors. Based on our study, a facial prosthesis exposed outdoors for approximately 8 to 12 h daily could have an equivalent service life of 2–3 years. The colour instability of nonpigmented silicone elastomers is one of the primary contributors to the overall colour degradation that occurs in service environments. Clinically observed discolouration of facial prostheses is a complicated phenomenon caused by several contributing factors, including the intrinsic colour stability of the silicone elastomer, pigments, flocking, loss of extrinsic colouration, personal habits (cleaning, cosmetics), and environmental factors (climate, fungus, and body oils) [32,33]. Colour deterioration in facial prostheses occurs because of UV-susceptible pigments and stains in various situations [23,34,35]. This may be due to post-polymerisation crosslinking caused by light irradiation [22], resulting in modifications to the polymer network structure. Initially, the number of polymer chains, their bonding, and their angular distribution in space was potentially modified, which possibly led to changes in the amount of light transmitted through the maxillofacial material as well as deterioration in the colour shade of the polymer. The A-2000 elastomer showed the highest colour stability in this study, which is in agreement with recent studies [23,36]. The current investigation evaluated the effect of intrinsic pigment Factor II on maxillofacial elastomers in outdoor weathering with respect to colour change only, which is a limitation of the present study. However, as previously reported for other dental materials, additional tests should be conducted to evaluate the effect of intrinsic pigment Factor II on flexural strength [37], hardness [38], and roughness [39] in outdoor weathering conditions. Nevertheless, maxillofacial prosthesis materials must have appropriate physical qualities, such as high tear resistance, high tensile strength, high modulus of elasticity, nonabsorption, and high wettability, besides colour stability [36,40]. When designing a maxillofacial prosthesis, it is essential to consider all these factors. Moreover, the climate of the Middle East is characterised by high temperatures, dryness, and intense sunlight that poses challenges to the durability and colour stability of maxillofacial silicones. The harsh environmental conditions, including high temperatures and exposure to UV radiation, can lead to changes in the mechanical properties and colour stability of silicone elastomers. However, limited research was conducted on the impact of the Middle East’s unique climatic conditions on the surface roughness and mechanical properties of different silicone elastomers. The resilience test, which measures a material’s ability to return to its former shape after deformation, is also important for understanding its elastic characteristics. Furthermore, studying the material’s fatigue resistance via cyclic loading testing would give vital insights into its long-term durability and performance under repetitive stress circumstances. Further studies should examine how other factors, such as sebaceous secretion storage, body temperature, and silicone cleaning solutions, affect the mechanical durability and colour stability of the tested silicone elastomers over a one-year period. Given the significance of these mechanical qualities for dental and maxillofacial materials, a thorough evaluation of the researched maxillofacial elastomers would include a mix of colour stability assessment and rigorous mechanical testing. A holistic approach such as this would give a more full knowledge of the material’s overall performance and appropriateness for usage in a demanding therapeutic setting.

## 5. Conclusions

Within the limitations of the study and the weathering conditions of the Al Jouf region, it can be concluded that after an extended period of six months of environmental exposure, the pigmented A-2000 RTV silicone showed better colour stability than the A-103 RTV silicone. Patients requiring facial prostheses will work outdoors, and weathering has deleterious effects on such prostheses.

## Figures and Tables

**Figure 1 materials-16-04331-f001:**
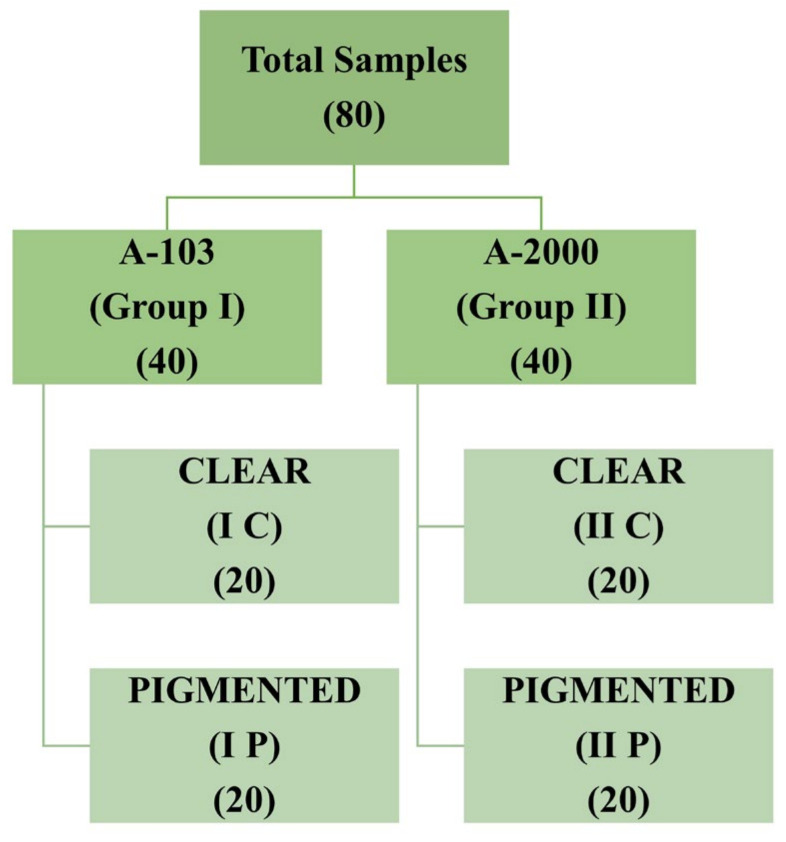
Group categorisation of prepared samples.

**Figure 2 materials-16-04331-f002:**
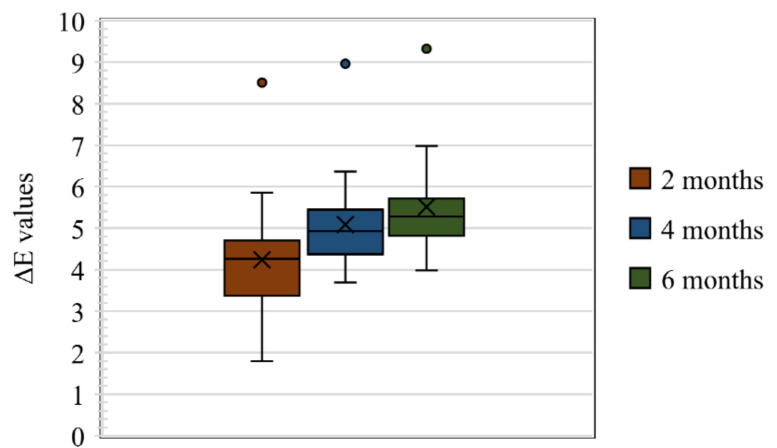
Box and whisker plot of ΔE values at 2, 4, and 6 months of Group I Control displaying range, median, first, and third quartiles.

**Figure 3 materials-16-04331-f003:**
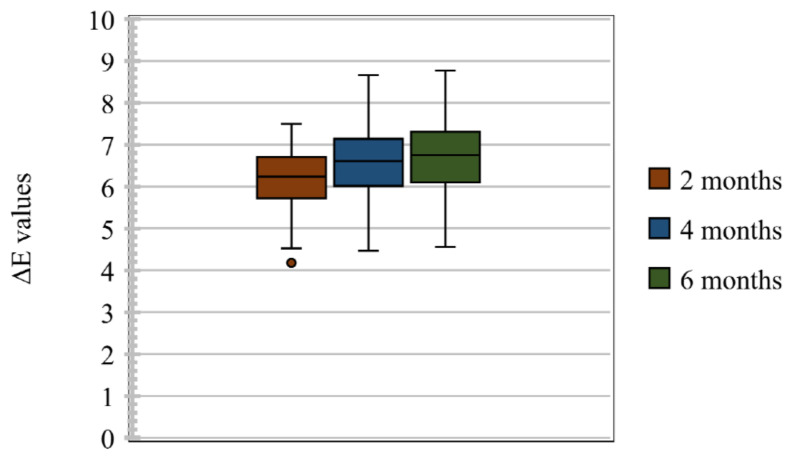
Box and whisker plot of ΔE values at 2, 4, and 6 months of Group I Pigmented displaying range, median, first, and third quartiles.

**Figure 4 materials-16-04331-f004:**
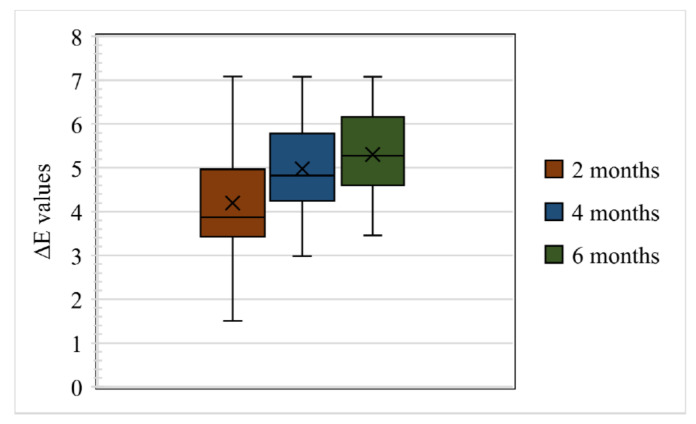
Box and whisker plot of ΔE values at 2, 4, and 6 months of Group II Control displaying range, median, first, and third quartiles.

**Figure 5 materials-16-04331-f005:**
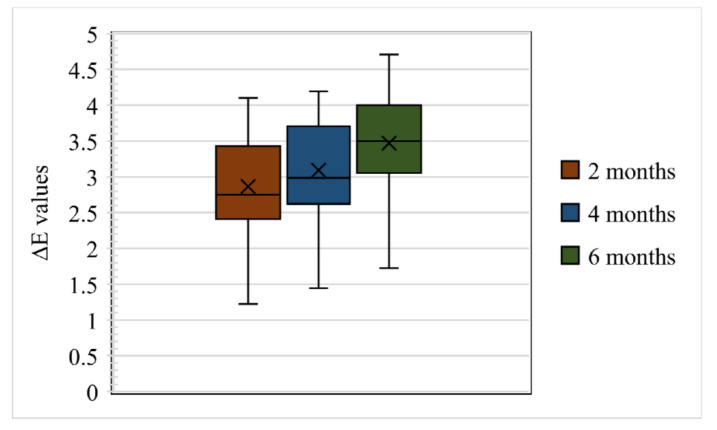
Box and whisker plot of ΔE values at 2, 4, and 6 months of Group II Pigmented displaying range, median, first, and third quartiles.

**Figure 6 materials-16-04331-f006:**
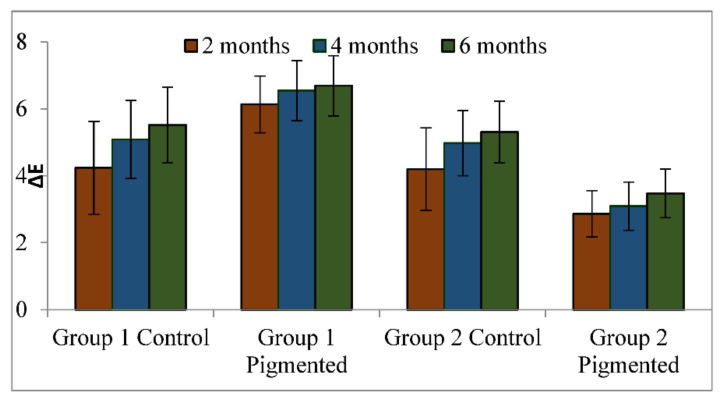
Comparison of mean values of ΔE in all the groups.

**Figure 7 materials-16-04331-f007:**
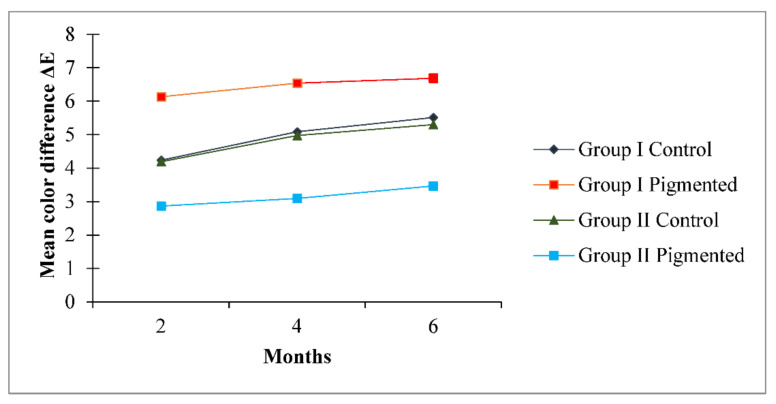
Comparison of mean values of ΔE over a period of 2, 4, and 6 months in all the groups.

**Table 1 materials-16-04331-t001:** Monthly average climatic data during outdoor weathering for Sakaka, Al Jouf, Saudi Araia.

Parameters/Month	October 2021	November 2021	December 2021	January 2022	February 2022	March 2022
Temperature (Max[daytime]–min[night time])	32 °C–19 °C	23 °C–13 °C	17 °C–9 °C	16 °C–7 °C	18 °C–9 °C	23 °C–12 °C
Monthly avg. wind speed km/h	12	13	12	13	14	15
Monthly avg. precipitation (mm)	10	15	12	12	8	8
Monthly avg. humiditiy (%)	25	39	46	46	38	30
Monthly avg. cloud cover	6%	12%	13%	14%	13%	11%
Monthly avg. UV index	6	5	4	4	5	5
Monthly avg. hours of sun	371	353	360	359	331	362

**Table 2 materials-16-04331-t002:** Descriptive statistics for the ΔE values in Group 1 Control, Group 1 Pigmented, Group II Control, and Group II Pigmented.

Groups	Time (Months)	N	Minimum	Maximum	Mean	Std. Deviation
Group I Control	2	20	1.80	8.51	4.23	1.39
	4	20	3.69	8.96	5.08	1.16
	6	20	3.98	9.32	5.51	1.12
Group I Pigmented	2	20	7.30	11.31	9.16	1.11
	4	20	7.64	11.43	9.54	1.06
	6	20	7.75	11.55	9.67	1.06
Group II Control	2	20	1.50	7.08	4.19	1.23
	4	20	2.98	7.08	4.97	0.97
	6	20	3.45	7.08	5.30	0.91
Group II Pigmented	2	20	1.22	4.10	2.86	0.69
	4	20	1.45	4.19	3.09	0.72
	6	20	1.73	4.71	3.47	0.72

**Table 3 materials-16-04331-t003:** Comparison of mean values of ΔE for all the groups.

ΔE	Groups	N	Mean	Std. Deviation (SD)	F Value	*p* Value
2 months	Group 1 Control	20	4.23	1.39	1.68	0.00
	Group 1 Pigmented	20	6.13	0.85		
	Group II Control	20	4.19	1.23		
	Group II Pigmented	20	2.86	0.69		
	Total	80	4.35	1.57		
4 months	Group 1 Control	20	5.08	1.16	0.46	0.00
	Group 1 Pigmented	20	6.54	0.89		
	Group II Control	20	4.97	0.97		
	Group II Pigmented	20	3.09	0.72		
	Total	80	4.92	1.54		
6 months	Group 1 Control	20	5.51	1.12	0.40	0.00
	Group 1 Pigmented	20	6.68	0.90		
	Group II Control	20	5.30	0.91		
	Group II Pigmented	20	3.47	0.72		
	Total	80	5.24	1.47		

## Data Availability

Data will be provided upon request by the corresponding author via email.

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
