# Peer review of "Colour Stability of Two Commercially Available Maxillofacial Prosthetic Elastomers after Outdoor Weathering in Al Jouf Province"

_materials, 2023, doi:10.3390/ma16124331_

Round 1
Reviewer 1 Report
The article presents interesting research on the effect of outdoor weathering on the colour stability of prosthetic elastomers. However, the manuscript needs improvement in line with the following comments:
- Subsections 2.1, 2.2 and 2.3 should be merged into one, also 2.4 and 2.4 (wrong number, page 3, line 125) can be merged into one section. Preparation of clear and pigmented samples;
- In line 115 the authors should change the description of the disc dimensions to 30mm diameter and 3 mm thick. The current notation is illegible.
- Authors should explain in the text what is part A and part B?
- Authors should add a caption to the figure on page 4 and reference it in the text.
- In line 157 the text "lightness coordinates. lightness, i.e," should be changed to "lightness coordinates, i.e. ..."
- In line 158 the text should be changed to: The values of L*, a*, and b* for each ...
- Tables 1 and 2 should not be divided into two pages;
- Authors should explain the parameters contained in the table: are the max and min temperatures the day and night temperatures? Are the wind speed and humidity values monthly averages? Are the precipitation values averages or sums of precipitation per month?
- In line 183, the authors should change the equation's form: ΔE = ([ΔL*]2 + [Δa*]2 + [Δb*]2)1/2, to a mathematical notation/form, symbols EXP and SQR are missing.
- Authors should provide the values of the parameters L*, a*, b* obtained from the tests in Table 2 and explain how they affect the resultant value of E. Does the increase of each parameter affect the increase of E?
- Captions of all figures should be placed under the figure;
- The table with values is unnecessary under figure 3,
- Authors should provide values up to 4 decimal places in table 2, in table 3 there are values for which zero before the decimal point is missing;
- In the discussion, the authors should comment on the lower test temperatures in relation to body temperature (36.6 degrees C). Will a higher temperature cause more changes?
I accept manuscripts after minor revisions.
Author Response
Respected Reviewer,
Thanks for providing your valuable comments. All the comments are being addressed using point to point clarification file attached herewith.
Regards

Reviewer 2 Report
Dear Authors,
Thank you for giving me the opportunity to study your work. I think the topic is interesting, your research is clinically relevant and tries to fill the gap in the scientific literature. Here are my suggestions about your paper:
- Lines 194-196: please rephrase properly
- Line 280: In my opinion the authors should change Prosthetics with Prostheses
- Line 285: please remove the article in “in the case of”
- Line 302: the first appearance of MFPs in the text should be explained and afterwards abbreviated
- Line 312: please use the plural prostheses
- Line 333: please use past tense
- Line 342: please change “the ” with “a”
- Line 389: please change with “requiring a facial prosthesis”
- Line 388-390: please rephrase properly, the meaning of the entire phrase is unclear
Author Response

(The authors gave the same response as above.)

Reviewer 3 Report
Dear Authors,
I have read the manuscript with interest and some questions raised. Enlisted please find my comments.
Overall. General English grammar revision (Minor spelling errors).
Key words. “dentistry” and “maxillofacial surgery” could be added in my opinion.
Abstract. Please add the names of the statistical tests in this section.
Introduction. Authors stated “External maxillofacial prosthesis is used for 57 several decades for anatomical, functional, or cosmetic restoration of the maxilla, mandible, or face that were missing or altered by disease, accident, or congenital malformation”. Please add a reference for this statement.
Materials and Methods. Authors stated “Master mold was fabricated using a disc of 30mm x 3mm of self-cure acrylic resin (Resin repair material, BMS dental, Italy) […]A hard wax (GEO Natural, Renfert, Germany)”. For each material used, please add details about commercial name manufacturer, City and State.
Materials and Methods. Authors stated “For preparation of 20 clear samples”. Please add if and how sample size calculation has been performed.
Materials and Methods. Authors stated “The vacuum mixing was carried out at 28 inches Hg for 30 minutes”. Please add a reference for this method.
Materials and Methods. Authors stated “The testing of sample was carried out on spectrophotometer (Spectro-shade micro II).”. For each machinery used, please add details about commercial name manufacturer, City and State.
Materials and Methods. Authors stated “Values were recorded in the CIELAB color system”. Please add a reference for this method.
Materials and Methods. Authors stated “Using the software SPSS (Version 22, IBM, NY) statistical analysis was done”. Please add details about software used, version, Manufacturer, City and State.
Materials and Methods. Authors stated “The data 189 for color difference were analyzed with one-way ANOVA (Analysis of variance)”. ANOVA is used for gaussian distributions. Please explain how normality of data was tested.
Materials and Methods. Authors stated “followed 190 by a post hoc test for identification of level of significance between every two groups”. Please add name of post hoc test.
Discussion. Authors stated “The A-RTV 2000 showed the most colour stability in this study in agreement with a recent study [26,32]”. Please add more discussion about other mechanical tests that could be conducted. Provide a general interpretation of the results in the context of other evidence, and implications for future research. It could be added that “The present report evaluated color changes. However, as previously reported for other dental materials, additional tests should be conducted to evaluate over time also flexural strength (Flexural strengths of fiber-reinforced composites polymerized with conventional light-curing and additional postcuring. Cacciafesta V, Sfondrini MF, Lena A, Scribante A, Vallittu PK, et al.. American Journal of Orthodontics and Dentofacial Orthopedics, 2007, 132(4), pp. 524–527), hardness (Hardness and Wear Resistance of Dental Biomedical Nanomaterials in a Humid Environment with Non-Stationary Temperatures. Pieniak D, Walczak A, Walczak M, Przystupa K, et al. Materials (Basel). 2020 Mar 10;13(5):1255) and roughness (Surface roughness of flowable resin composites eroded by acidic and alcoholic drinks. Poggio, C., Dagna, A., Chiesa, M., Colombo, M., et al. Journal of Conservative Dentistry, 2012, 15(2), pp. 137–140) also for maxillofacial elastomers, in order to complete the overview about their characteristics”. These concerns should be added to Discussion section.
Discussion. Please add a paragraph showing the limitations of the present report.
References. Some references are quite old (1980;1990;1980). If possible, please switch with some more modern research. Some recent studies have been suggested in the sections above.
Table2. For each variable tested, please add a table with descriptive statistics (mean, standard deviation, minimum, median and maximum values) of the various groups.
Figure 6. Please add standard deviations or other dispersion measure.
Author Response

(The authors gave the same response as above.)

Round 2
Reviewer 3 Report
Dear Authors,
I have read the revised manuscript but surprisingly many comments remained ignored. Enlisted please find my comments again.
Overall. General English grammar revision (Minor spelling errors).
Key words. “dentistry” and “maxillofacial surgery” could be added in my opinion.
Abstract. Please add the names of the statistical tests in this section. Post hoc tests are lacking.
Materials and Methods. Authors stated “Master mold was fabricated using a disc of 30mm x 3mm of self-cure acrylic resin (Resin repair material, BMS dental, Italy) […]A hard wax (GEO Natural, Renfert, Germany)”. For each material used, please add details about commercial name manufacturer, City and State. Cities of manufacturers are still lacking.
Materials and Methods. Authors stated “For preparation of 20 clear samples”. Please add if and how sample size calculation has been performed. Please add details about primary outcome.
Materials and Methods. Authors stated “The testing of sample was carried out on spectrophotometer (Spectro-shade micro II).”. For each machinery used, please add details about commercial name manufacturer, City and State. City of manufacturer is still lacking.
Materials and Methods. Authors stated “Using the software SPSS (Version 22, IBM, NY) statistical analysis was done”. Please add details about software used, version, Manufacturer, City and State. City of manufacturer is still lacking.
Discussion. Authors stated “However, maxillofacial prosthesis 373
materials must have appropriate physical qualities such as high tear resistance, high tensile strength, high modulus of elasticity, nonabsorption, and high wettability, besides colour stability [35, 36]”. This is incomplete. Please add more discussion about other mechanical tests that could be conducted, as per other dental and maxillofacial materials. Provide a general interpretation of the results in the context of other evidence, and implications for future research. It could be added that “The present report evaluated color changes. However, as previously reported for other dental materials, additional tests should be conducted to evaluate over time also flexural strength (Flexural strengths of fiber-reinforced composites polymerized with conventional light-curing and additional postcuring. Cacciafesta V, Sfondrini MF, Lena A, Scribante A, Vallittu PK, et al.. American Journal of Orthodontics and Dentofacial Orthopedics, 2007, 132(4), pp. 524–527), hardness (Hardness and Wear Resistance of Dental Biomedical Nanomaterials in a Humid Environment with Non-Stationary Temperatures. Pieniak D, Walczak A, Walczak M, Przystupa K, et al. Materials (Basel). 2020 Mar 10;13(5):1255) and roughness (Surface roughness of flowable resin composites eroded by acidic and alcoholic drinks. Poggio, C., Dagna, A., Chiesa, M., Colombo, M., et al. Journal of Conservative Dentistry, 2012, 15(2), pp. 137–140) also for maxillofacial elastomers, in order to complete the overview about their characteristics”. These concerns should be added to Discussion section.
Discussion. Please add a paragraph showing the limitations of the present report.
References. Some references are still old (2002). If possible, please switch with some more modern research. Some recent pertinent studies have been suggested in the sections above.
Please take into careful account all the considerations. Thank you.
Author Response
Respected Reviewer,
- We appreciate your insightful comments and suggestions for improving our work.
- Your feedback has greatly enhanced the quality and clarity of our manuscript.
- We are grateful for your constructive criticism, which has helped us to refine our discussion and findings.
All the comments are addressed using point to point clarification file attached herewith.
Thanking you
Regards

Round 3
Reviewer 3 Report
All comments have been assessed, thank you.
Author Response
Dear Reviewer,
Your comments have helped us to improve the clarity and coherence of our manuscript, and we are grateful for your guidance.
Thank you very much
Regards
Dr Kiran Ganji